# The Potential of Naturally Derived Compounds for Treating Chronic Kidney Disease: A Review of Autophagy and Cellular Senescence

**DOI:** 10.3390/ijms26010003

**Published:** 2024-12-24

**Authors:** Yoong Mond Teh, Siti Aisyah Mualif, Nur Izzati Mohd Noh, Soo Kun Lim

**Affiliations:** 1Department of Biomedical Engineering and Health Science, Faculty of Electrical Engineering, University Technology Malaysia (UTM), Johor Bahru 81310, Malaysia; yoongmond@graduate.utm.my (Y.M.T.); aisyahmualif@utm.my (S.A.M.); 2Department of Biosciences, Faculty of Science, University Technology Malaysia (UTM), Johor Bahru 81310, Malaysia; izzati@utm.my; 3Department of Medicine, Faculty of Medicine, University of Malaysia (UM), Kuala Lumpur 59100, Malaysia

**Keywords:** chronic kidney disease, renoprotective, autophagy, cellular senescence, natural product, traditional Chinese medicine

## Abstract

Chronic kidney disease (CKD) is characterized by irreversible progressive worsening of kidney function leading to kidney failure. CKD is viewed as a clinical model of premature aging and to date, there is no treatment to reverse kidney damage. The well-established treatment for CKD aims to control factors that may aggravate kidney progression and to provide kidney protection effects to delay the progression of kidney disease. As an alternative, Traditional Chinese Medicine (TCM) has been shown to have fewer adverse effects for CKD patients. However, there is a lack of clinical and molecular studies investigating the mechanisms by which natural products used in TCM can improve CKD. In recent years, autophagy and cellular senescence have been identified as key contributors to aging and age-related diseases. Exploring the potential of natural products in TCM to target these processes in CKD patients could slow disease progression. A better understanding of the characteristics of these natural products and their effects on autophagy and cellular senescence through clinical studies, coupled with the use of these products as complementary therapy alongside mainstream treatment, may maximize therapeutic benefits and minimize adverse effects for CKD patients. While promising, there is currently a lack of thorough research on the potential synergistic effects of these natural products. This review examines the use of natural products in TCM as an alternative treatment for CKD and discusses their active ingredients in terms of renoprotection, autophagy, and cellular senescence.

## 1. Traditional Chinese Medicine in CKD: Aging and Senescence Focus

Chronic kidney disease (CKD) is typically defined as the gradual loss of kidney function, and is a progressive condition characterized by structural and functional changes to the kidney [1]. CKD is a significant global health concern, currently affecting approximately 10% of the global population. This statistic underscores the widespread prevalence of the disease [2]. Glomerular disease, including nephrotic syndrome, is one of the leading causes of end-stage kidney disease (ESKD). ESKD is the last stage of long-term chronic kidney disease. Diabetes mellitus, minimal change disease, focal segmental glomerulosclerosis (FSGS), and membranous nephropathy are the common causes of nephrotic syndrome [3]. Proteinuria is one of the main symptoms of glomerular disease [3], a condition of protein (predominantly albumin) found in the urine, which is directly caused by podocyte injury. When podocyte foot processes enlarge and lose their integrity, they shape to maintain the slit diaphragm, which has a critical role in preventing the leak of plasma protein into the urine, and hence cause proteinuria [4]. Corticosteroids and immunosuppressive medication are the mainstream therapies for glomerular disease. The gruesome side effects from these mainstream therapies compromise the patient’s health and sometimes even worsen the condition. It is common for glomerular disease patients from China and other Asian countries [5], where the local authority formally accepts Traditional Chinese Medicine (TCM) as the dual health system [6,7], to be willing to try TCM because TCM offers an alternative to patients. In nephrotic syndrome, those who are steroid-resistant may progress to CKD and ESKD. TCM might be an adjuvant therapy, and might be a chance for recovery or slowing down the CKD progression [8] because TCM is an effective adjuvant therapy that improves long-term survival in patients with CKD [9]. Another way of looking at TCM’s role in modern medicine is by using natural products as a prescription treatment.

Natural products are inexpensive, readily available, and associated with fewer adverse effects [10] compared to modern medicine. The use of natural products to prevent and treat CKD is gaining attention in Europe, North America, and Asia [10]. However, most TCM applications in CKD have not explored the aspect of aging and cellular senescence or prognosis to investigate aging and cellular senescence at the molecular level in CKD. Other challenges in studying Traditional Chinese Medicine (TCM) for CKD include the limited availability of English-language studies, formal data documentation, lack of properly designed systematic studies, preclinical studies, clinical studies, and randomized trials. Treatment recommendations are often primarily based on the experience of TCM practitioners [5]. Nonetheless, solid clinical studies have begun to confirm and uncover emerging scientific evidence supporting the mechanisms of TCM [11]. Additionally, certain herbs and ingredients, often passed down by word of mouth, are believed to have kidney-protective effects, which patients may readily adopt in the hope of recovery. Researchers, meanwhile, are increasingly interested in the molecules involved in these protective mechanisms. This review aims to highlight the scientific evidence for kidney-protective effects from natural substances used in TCM, while also providing a perspective on the connections between aging, autophagy, cellular senescence, and these natural substances. The potential involvement of therapeutic targets at the molecular level not only from TCM, but also from other natural products, is discussed, to offer researchers direction for further investigation into CKD pathogenesis. Understanding the interplay between aging, cellular senescence, and CKD is crucial to fully harness the potential of naturally derived compounds in treating kidney diseases.

## 2. Aging and Cellular Senescence

Aging has been recently recognized as a disease and also a significant contributor to increase acute kidney disease (AKD) and chronic kidney disease (CKD) [12]; kidney aging pathogenesis includes the activation of various injury pathways, a decrease in podocyte autophagy, the release of cytokines and growth factors, and an increase in reactive oxygen species (ROS) and advanced glycation end products (AGEs) [13]. Signaling pathways reviewed by Teh et al. provided insight into aging-related autophagy-mTOR in podocyte injury, which is highly associated with foot process effacement (podocyte injury), which causes proteinuria [14]. It is believed that delaying aging is much more effective than treating a particular age-related disease. Figure 1 shows that cellular senescence contributes to kidney aging through decreased autophagy, loss of nephrons, and activation of injury pathways via senescence-associated secretory phenotype (SASP), while DNA damage, together with stress-induced senescence, oncogene-induced senescence and replicative senescence further drive chronic and acute pathological processes.

Cellular senescence (CS) is a fundamental feature that manifests and aggravates the condition of aging [13]. CS is an irreversible cell cycle arrest in which the cell still has metabolic activity, but no capability of replication [15]. There are three broad ranges of CS that were defined based on induction: replicative senescence (telomerase attrition), oncogene-induced senescence (activation of oncogenes), and oxidative stress-induced senescence [16]. A series of proteins secreted by the senescent cells is known as the senescence-associated secretory phenotype (SASP), an essential characteristic of senescent cells. The SASP will trigger the adjacent cells to release various inflammatory factors such as cytokines, growth factors, chemokines, and extracellular matrix remodeling factors [15]. The key regulator of SASP is known as GATA4, a transcription factor regulated by autophagy, but once the cell experiences senescence-inducing stimuli, it will escape the regulation by autophagy and start to accumulate [17]. However, for the kidney, cellular senescence was described by Schmitt et al. as referring to the intrinsic cellular responses to cumulative, low-level damage that occurs over the course of an individual’s lifespan and which may not become apparent until a certain threshold is reached [12]. On the other hand, extracellular vesicles (EVs) play dual roles in cellular senescence with SASP in chronic kidney disease, and are currently receiving attention from researchers in focusing on its involvement in kidney repair and a therapeutic direction [18]. The details of the role of EVs was examined in [18]. Therefore, exploring the signaling pathways of CS, autophagy, and SASP in CKD can provide insights into novel therapeutic targets.

## 3. The Role of SASP in Aging, Autophagy, and Podocyte

DNA damage response (DDR) illustrated in Figure 2 is regulated by ATM and ATR (ATM- and Rad3-Related) kinases [19]. Initial tissue repair in response to DNA strand breaks is mediated by ataxia–telangiectasia mutated (ATM) activation and histone H2AX phosphorylation [20]. When this initial response fails, DDR is intensified, leading to the activation of transcription factor NF-κB and CCAAT enhancer binding protein β (C/EBP-β), which subsequently induces the production of various SASP factors. Notably, NF-κB can also be activated by p38/mitogen-activated protein kinase (MAPK) pathway, independent of persistent DNA damage repair [21]. The SASP shares similarities with CKD-associated secretory phenotypes, suggesting a crosstalk between CS and CKD. In terms of CS’s beneficial role in the kidney, recent findings suggest that acute senescence, marked by a transient SASP presence, is a programmed short-duration response to specific stressors that promotes renal regeneration, restricts renal fibrosis, and supports wound healing following injury [22]. In contrast, chronic senescence, characterized by prolonged SASP presence, ineffective clearance of senescent cells, long-term cellular oxidative stress, macromolecular damage, and the accumulation of senescence cells, is detrimental to renal health. This chronic state contributes to renal aging and the irreversible decline of kidney function [22].

The roles and mechanisms of the sirtuin family in DNA damage repair and response have been reviewed by Lagunas-Rangel et al. [23] and sirtuin’s role in podocyte autophagy was recently examined [14,24]. Cooper et al. reported that sufficient DNA damage would induce SASP development. Sirtuin is crucial in regulating genome integrity by maintaining the chromatin condensed state and responding to DNA damages and repairs [25]. Sirtuin is essential in metabolism signaling pathways that delay CS, extending the organism’s lifespan through various cellular processes. Lee et al. had recently reviewed the role of sirtuin in cellular senescence and aging [25]; however, Vazquez et al. discussed the role of sirtuin 7 more deeply [26]. Both reviews highlighted the fact that DNA damage and repairs both respond to sirtuin 1, sirtuin 6, and sirtuin 7. Nonetheless, the recruitment of sirtuin towards the DNA damage sites provided an important clue with regard to sirtuin 1, 6 and 7. Sirtuin 1 and 6 responded in a fast kinetic manner towards the DNA damage sites to initiate the damage response, while sirtuin 7 was recruited with much slower kinetics, providing insight into its distal role in the DNA damage repair process [26]. Sirtuin 7 is recruited for DNA damage repair by deacetylating histone H3 at lysine 18, which has an exact role in facilitating the recruitment of protein 53BP1 to the DNA damage sites [26]. Investigating natural products from the perspectives of sirtuin could provide a deeper understanding of their mechanisms in CKD treatment.

Autophagy is an essential catabolic process for cellular self-cleansing, responsible for degrading damaged and unhealthy intracellular materials. Across various model organisms, autophagy’s beneficial role in the age-association processes and lifespan determination has been indicated. Recent studies revealed concrete evidence to establish the role of autophagy in aging, particularly a crucial factor involved in podocyte injury [14]. GATA4 is a transcription factor that serves as a key regulator of the SASP and cellular senescence. Interestingly, autophagy can both activate and inhibit CS through the autophagic receptor protein sequestome 1 (SQSTM1/p62), which mediates GATA4 degradation under homeostasis conditions [17]. However, the interaction between SQSTM1/p62 and GATA4 is disrupted when the cell experiences senescence-induced stimuli; GATA4 escapes autophagic inhibition, begins to accumulate, and this accumulation leads to SASP activation via the NF-kB transcription pathway [17]. The relationship between autophagy and CS is complex, with speculation that autophagy modulates several targets that oppositely regulate CS; hence, it has been suggested that varying autophagy conditions could yield different outcomes [17]. Figure 3 illustrates the direct interplay between key regulators of autophagy and cellular senescence, highlighting their influence on the progression and condition of CKD. Podocytes exhibit high autophagic activities, relying on autophagy for “self-cleaning”. Contrary to the typical aging perspective that autophagy activity decreases with cellular aging, Schmitt et al. (2017) found that autophagy activity in aged podocytes is upregulated. This observation has led to speculation about a fundamental distinction between baseline autophagy and stress-induced autophagy [12], or about the fact that autophagic activities might decrease due to lysosomal dysfunction and/ or abnormalities in autophagosome maturation [24]. Selective autophagy is believed to be regulated by ATM and ATR kinases during DDR. According to Kang et al. (2016), general autophagy supports the transition to senescence, whereas selective autophagy suppresses CS by degrading the senescence regulator (GATA4) [17].

Podocytes, fundamental units of the kidney, are terminally differentiated cells with the highest autophagic activity in the kidney [27]. Autophagy is essential for cellular housekeeping in cells such as neurons and podocytes [27]. It is a self-digestion process that facilitates the massive degradation of cellular proteins [28]. Autophagy in kidney cells may serve as a protective mechanism that slows the progression of chronic kidney disease (CKD) [29], particularly associated with proteinuria and podocyte foot process effacement in Minimal Change Disease [28]. However, a distinction may exist between baseline autophagy and stress-induced autophagy in podocytes [12]. This hypothesis arises from observations in aged mice, where baseline autophagy did not decline with age, yet autophagy levels did not increase in response to starvation. This raises questions regarding the threshold of autophagic activity required for specific cellular conditions. The autophagic receptor protein SQSTM1/p62 degrades GATA4, a regulator of the senescence-associated secretory phenotype (SASP) and cellular senescence, underscoring autophagy’s role in mediating age-related diseases.

Global glomerulosclerosis is a critical hallmark of glomerular aging. The senescent cells that secrete the senescence-associated secretory phenotype (SASP) are primarily various glomerular cell types, especially the terminally differentiated cells such as podocytes. Senescent podocytes release profibrogenic SASP signals through the paracrine mechanism, influencing neighboring cells and contributing to the development of glomerulosclerosis [30]. The cell cycle and senescence of podocytes have been reported to be regulated by podocyte Histone Deacetylase 1 and 2 (Hdac1 and 2) [31]. The development of senescent cells depends on the cell type and organ. Renal p16INK4a expression has been proposed as the hallmark for assessing kidney aging [12,21]. Podocytes, as terminally differentiated cells, possess limited proliferative and regenerative capacities, but can recover from stress and injuries. However, podocyte health is particularly vulnerable when affected by neighboring senescent cells, leading to a gradual loss of surrounding cells. The loss of podocyte (resulting in podocyte hypertrophy) increases the burden on the remaining podocytes, which must endure greater stress. This compensatory workload is sustained over time, and as kidney aging progresses, the kidney’s capacity to recover from injuries is further compromised [12]. The signaling pathways of senescent cells vary across different cell types, yet little is known about renal SASP signaling [32]. Investigating the signaling pathways linking senescent podocyte with autophagy is crucial for developing improved therapeutic strategies for the future.

Conversely, lithium has been used as a GSK3β inhibitor to enhance the Wnt signaling pathway, which may negatively impact cellular senescence, as Wnt9a upregulates senescence markers such as p53, p16, and p21 [33]. While GSK3β and Wnt may regulate each other in specific contexts, studying the role of sirtuin within this interaction is crucial, as sirtuin is directly involved in autophagy and mTOR pathways. However, the interactions between sirtuin, Wnt, and GSK3β remain largely unexplored. Given GSK3β’s broad involvement in cellular senescence and various human diseases, significant gaps remain in understanding its precise mechanisms within the autophagy signaling pathway and the specific role of the Wnt signaling pathway in cellular senescence. It is possible that GSK3β primarily impacts β-catenin, which is essential for podocyte homeostasis [30]. The Wnt/β-catenin signaling pathway has been found to contribute to kidney aging, and studies suggest that klotho may inhibit this pathway to prevent cellular senescence [34]. Interestingly, the role of lithium in cellular senescence and its potential longevity effects underscore the need for further research into the relationships among lithium, GSK3β, and aging, which could offer promising therapeutic strategies for age-related diseases. Currently, the roles of GSK3 in cellular senescence, autophagy, and chronic kidney disease (CKD) remain largely unknown.

Recently, some evidence suggests glycogen synthase kinase 3 (GSK3) plays a key role in cellular senescence (CS) [22] and autophagy by inhibiting the mTOR pathway through phosphorylation of TSC2 [14]. For many years, GSK3β was the inhibitor target for the canonical Wnt signaling pathway [35]. GSK3 acts as a converging point for multiple signaling pathways such as inflammation, tissue injury, repair, and regeneration. It was reported that lithium-induced inhibition of GSK3 extends the organism’s lifespan [36]. Another interesting fact is that, based on large-scale population-based studies, the level of lithium in drinking water significantly correlates with longevity [37]. A large Japanese cohort study involving 1,206,174 individuals across 18 municipalities demonstrated a significant inverse correlation between lithium levels in drinking water and all-cause mortality. Similarly, low-dose lithium chloride exposure in Caenorhabditis elegans was shown to extend lifespan (*p* = 0.047). These findings highlight the potential conserved anti-aging properties of long-term, low-dose lithium exposure across species, emphasizing the need for further investigation into its potential benefits for human longevity [37]. Another piece of compelling evidence for glomerular aging was that GSK3β regulates the senescence signaling pathway for glomerular podocytes, and biochemically, it has been the primary well-known molecular target of lithium [30,36]. Bao et al. validated lithium as an autophagy inducer and suggested that it activates autophagy via AMPKα phosphorylation [38]. Understanding CS and autophagy open avenues for exploring systemic factors that influence aging and CKD, such as the gut microbiome.

## 4. The Role of the Gut Microbiome in Longevity

The gut microbiome is rapidly emerging as a critical regulator of various aspects of human health, with an increasingly recognized relationship to CS. CKD is often associated with gut dysbiosis [39] and the accumulation of toxic metabolites, which reportedly interact with the intestinal barrier to influence the heart and kidneys, forming a Gut–Heart–Kidney crosstalk [40]. Altered gut microbiome composition has been reported to affect SASP-mediated inflammation and immunosenescence. Furthermore, specific bacterial metabolites have been shown to suppress premature, stress-induced CS and SASP, highlighting the gut microbiome’s role in preventing CS [16]. An intriguing study by Siddiqui et al. explored the longevity of crocodiles, suggesting a link between CS and their unique gut microbiome composition [41]. Crocodile, often referred to as “living fossils”, have existed for the past 85 million years and have survived numerous catastrophic events. Studies indicate that crocodiles’ remarkable resilience, limited CS, adaptability to environmental changes, and resistance to infections are likely influenced by their gut microbiome [41].

The composition of the gut microbiome is associated with various human aging conditions, including the accumulation of aged immune cells. Moreover, aging has been linked to decreased diversity in the gut microbiome, particularly a reduced Firmicutes-to-Bacteroides ratio [41]. Studies reveal that animals with exceptionally long lifespans, such as crocodiles and desert tortoises, exhibit limited and negligible signs of senescence. The genome of the crocodile has evolved very slowly over millions of years [41]. This observation presents an optimistic and encouraging avenue for studying human and crocodile gut microbiomes to uncover new insights. These findings underscore the potential of targeting the gut microbiome and cellular senescence (CS) pathways, possibly with natural products used in traditional Chinese Medicine (TCM), for the treatment of chronic kidney disease (CKD).

The role of the gut microbiome in CKD, aging, and various age-related diseases remains largely unexplored. Few in vitro or in vivo studies have investigated how the gut microbiome affects CKD. However, increased attention to the gut microbiome in CKD research has highlighted the impact of gut-derived uremic toxins, such as p-cresyl sulfate (PCS), indoxyl sulfate (IS), and trimethylamine N-oxide (TMAO), on CKD progression, although the in-depth relationships remain largely unknown [42]. Gut dysbiosis is commonly observed in CKD patients [43]. Recent scientific findings suggest that factors such as diet [44], sleeping cycle [45], exercise, circadian rhythm, and lifestyle—all through the lens of aging—have gained attention for their potential to intervene in idiopathic age-related diseases. This approach could help treat or prevent these diseases, improve patient survival and well-being, and deepen understanding of specific disease pathogenesis. The gut microbiota, as the largest micro-ecosystem in the human body, produces metabolites that are closely linked to chronic metabolic diseases. Non-invasive microbiome biomarkers for CKD have been identified in different regions of China, as microbiome species can vary by location [42]. The intriguing discovery of the gut microbiome in crocodiles has further underscored its potential role in aging, cellular senescence, and various age-related diseases. Understanding the mechanisms and relationships between the gut microbiome, CKD, and aging presents challenges due to biological complexity, multifactorial influences, and individual variability. Currently, the knowledge and experimental models needed to explore these aspects are limited. However, rapid advancements in genetic screening and DNA/RNA analysis may help manage the large data sets involved. Machine learning technology may further support precision and yield novel insights into gut microbiome studies [46].

## 5. Natural Products Used in TCM for CKD

Curcumin is a lead compound with potential for treating and preventing nephrotic disease [47]. Its anti-inflammatory effects improve the bioavailability of nitric oxide and scavenge free radicals, specifically reactive oxygen species, due to the presence of phenolic compounds [48,49]. Curcumin’s antioxidant properties are bifunctional: it not only directly reacts with reactive oxygen species, but also triggers the upregulation of various cytoprotective and antioxidant proteins, such as Nrf2 [49,50]. Curcumin has been extensively studied for its health benefits, and numerous clinical trials have been conducted. However, the most prominent issue with curcumin is its low bioavailability [49]. Although curcumin has been a significant research focus, studies examining the specific dosages required for chronic kidney disease (CKD) patients remain limited [49]. Curcumin may play a crucial role in improving kidney function, protecting podocytes [47], improving blood pressure by suppressing Renin–Angiotensin system receptor expression [51] and protecting the kidney from renal failure [52]. Despite curcumin being one of the most extensively researched natural ingredients included in this review, to the best knowledge of the authors, there are no direct studies comparing its efficacy against current mainstream CKD medications, such as SGLT2 inhibitors. Furthermore, in double-blind studies investigating prednisolone-induced insulin resistance and other glucometabolic disturbances, curcumin showed no protective effect [53]. Other natural products have also shown promise in modulating autophagy and cellular senescence (CS) pathways relevant to CKD.

Curcumin’s role in both autophagy and cellular senescence positions it as a potent therapeutic candidate for age-related diseases and aging. However, despite the potential benefits of curcumin, some studies have reported that while it reduced oxidative stress, it did not improve proteinuria [48]. A study by Avila-Rojas et al. suggested that curcumin’s nephroprotective effects may be related to its ability to restore autophagy flux balance and regulate autophagic flow [50]. In the context of cellular senescence, although curcumin can evoke a broad range of cellular responses to external stimuli and interact simultaneously with multiple receptors, growth factors, transcription factors, kinases, enzymes, and other proteins, it does not inhibit or postpone cellular senescence, and might even induce it. It has been suggested that curcumin may instead promote changes in proteins involved in aging or cellular senescence through the induction of sirtuin and AMPK pathways [54]. As of the date of this review, the clinical effects of curcumin have primarily been studied in cancer patients, with no clinical applications reported for CKD patients [55].

Oxidative stress plays a vital role in the pathogenesis of chronic kidney disease (CKD). The accumulation of advanced oxidation protein products (AOPPs) has a direct effect on podocyte injury, proteinuria, and the development of focal segmental glomerulosclerosis (FSGS) [56,57]. Epigallocatechin-3-gallate (EGCG), the major catechin in green tea extract, is one of the most potent antioxidants against oxidative stress and acts as a scavenger of reactive oxygen species [58]. It has been reported to have renoprotective effects in animal studies [59]. Unfermented green tea contains the highest concentration of catechins (50% EGCG) compared to other fermented teas, such as white, yellow, and black teas [58]. Despite various studies demonstrating the potential benefits of EGCG [59], clinical research on its effects in CKD remains limited. EGCG is inherently unstable and undergoes auto-oxidation under cell culture conditions, which could impede research progress. Nevertheless, EGCG continues to be of significant research interest for its potential in attenuating oxidative stress and inflammatory responses [60].

A few studies have identified astragaloside IV (AS-IV), a chemical component of *Astragalus membranaceus*, as an active compound for reducing oxidative stress [61]. Table 1 summarizes the natural products and active ingredients that have been investigated for chronic kidney disease (CKD), while Table 2 highlights the autophagic characteristics of natural products that show promising therapeutic effects on CKD. Additionally, its effect on reducing proteinuria has been reported in several Chinese-language journals [5]. *Astragalus membranaceus* has been shown to enhance autophagy by activating the AMPK-mTOR pathway [62]. The use of cyclophosphamide in patients with nephrotic syndrome can lead to the production of oxygen free radicals, which contribute to oxidative stress. Consequently, Astragalus, as a plant with high antioxidant content, is particularly useful for mitigating oxidative stress [63]. Moreover, astragaloside IV has been reported to inhibit senescent cell development via mitophagy, making it a potent protective compound [64]. An animal study demonstrated that astragaloside IV may delay aging by regulating the sirtuin 1/p53 signaling pathway [65]. Based on these findings, Astragalus has been recommended for co-prescription with cyclophosphamide [63]. The beneficial effects of Astragalus have been consistently demonstrated in numerous clinical studies, showing efficacy in treating nephrotic syndrome [66]. The Tangshen formula, an extract comprising Astragalus and six other natural herbs, has also been shown to increase sirtuin 1 expression and promote autophagy [67]. A longevity-related transmembrane protein with anti-aging features mainly secreted primarily by the kidney is known as Klotho. Klotho is a major modulator of CS that regulates the p53/p21 signaling pathway. Astragaloside was reported to be able to upregulate Klotho expression [68], and protects podocytes against apoptosis by inhibiting oxidative stress by activating the PPARγ-Klotho-FoxO1 signaling pathway [69]. Rhein has been shown to prevent the suppression of Klotho that occurs as a result of acute inflammation. Furthermore, the reversal of Klotho suppression by Rhein has been shown to mitigate acute inflammation and kidney injury induced by LPS [70].

Resveratrol, a non-flavonoid natural polyphenol, has been investigated for its potent antioxidant properties, with studies indicating that it can reduce antibody production (IgG and IgA) [71]. While the exact mechanisms of resveratrol’s effects remain unclear, research has consistently highlighted its antioxidant, anti-inflammatory, and renoprotective features [72] and its involvement in several signaling pathways, including SIRT1 [71] and the STAT pathway, which reduces inflammatory cytokines [73]. In studies on hypertensive injury in rats, resveratrol demonstrated beneficial effects, primarily attributed to activation of the Nrf2 signaling pathway [72]. The production and accumulation of advanced glycation end-products (AGEs) can induce specific renal lesions [74]. Resveratrol may interfere with the effects of AGEs, such as oxidative stress and NF-κB activation, which further enhances ROS production [29]. Both EGCG and resveratrol have been reported to ameliorate the development of premature senescent cells [16]. Furthermore, both compounds have been shown to mimic calorie restriction effects, potentially modulating organism lifespan [16].

Triptolide, the principal active ingredient from Tripterygium Wildfordii Hook F [75] has been shown to be beneficial when combined with cyclosporine A therapy [75]. A very low dose of cyclosporine A, used in combination with triptolide, can protect podocytes from injury and preserve cytoskeletal integrity [75]. However, triptolide administration may impair the kidney’s antioxidant system, potentially causing nephrotoxicity [76]. Additionally, triptolide has been reported to suppress CD80 and CD86 [77]. Due to its CD80-suppressing ability, triptolide plays an important role in rheumatoid arthritis treatment [78], and CD80 expression is linked to the frequent relapse of the group of minimal change diseases, a form of chronic kidney disease [79]. In the context of cellular senescence, triptolide has been observed to increase p53 and p21 levels, which can arrest the cell cycle and promote cellular senescence [80,81]. Further clinical studies are needed to fully assess the therapeutic potential of triptolide. A low dose of cyclosporin A combined with triptolide may prevent kidney injury; however, triptolide alone may cause podocyte injury. Given findings that triptolide can induce cellular senescence (CS) and may contribute to kidney injury, further studies are needed to determine whether triptolide-induced CS and kidney injury are transient or chronic.

Tripterygium root, when used in combination with Cordyceps, has been found effective in attenuating glomerular damage by preventing the downregulation of podocyte-expressed proteins such as podocin and nephrin [82]. Cordyceps cicadae has medicinal benefits similar to *Cordyceps sinensis* and *Cordyceps militaris* [83]; however, further research and standardized benchmarking methods are required to provide more definitive clinical evidence regarding the benefits of its bioactive compounds. This fungus has long been valued in China as a kidney tonic, and it has been reported that Cordyceps can reduce cyclosporine-induced nephrotoxicity [84]. Wei Yan et al. demonstrated that Cordyceps cicadae has protective effects against oxidative stress-induced cellular senescence [85]. *Cordyceps sinensis* was concluded by Wang et al. to alleviate stress-induced cellular senescence [86] and has been reported to exert significant nephroprotective effects [87]. Additionally, derivatives of Cordyceps, specifically cordycepin, have been shown to inhibit cellular senescence via the NRF2 and AMPK signaling pathways [88].

The antioxidant properties of emodin, primarily derived from *Rheum palmatum* (rhubarb), Cassia occidentalis, and Aloe vera [73] have been extensively studied in China for their benefits for chronic kidney disease (CKD). It has been found that rhubarb can lower serum creatinine levels and reduce urinary protein excretion [89], both of which are critical concerns for CKD patients. *Rheum palmatum* contains numerous active compounds with potential therapeutic effects on CKD, among which anthraquinones rhein and emodin are the primary active ingredients. Various preclinical models [90] and clinical studies have shown that rhubarb can improve renal function and reduce proteinuria [5]. Low-dose emodin has been reported to induce cellular senescence, potentially preventing breast cancer development. This low dose of emodin may also enhance sensitivity to chemotherapy through cellular senescence, with emodin-induced senescence being potentially reversible [91].

*Schisandra sphenanthera* (Wuzhi) has shown significant efficacy when co-administered with tacrolimus [92] and low dosage of cyclosporine [93] in rat studies. However, the outcomes observed in animal models should be further investigated in clinical studies. Wuzhi, a prescribed medication in China, is commonly used for liver protection and in liver and kidney transplantation [92]. It has been approved for co-administration with tacrolimus in China [66]. Studies have indicated that Wuzhi exhibits biphasic effects, suggesting that an optimal dosage should be determined to achieve beneficial outcomes when co-administered with different medications [92]. A species similar to *Schisandra sphenanthera*, known as Schisandra chinensis, shares the same appearance and can only be distinguished through microscopic examination. Schisandra chinensis has been associated with autophagy, alleviation of oxidative stress, suppression of stress-induced premature senescence, and reduction in proinflammatory markers [94,95]. Recently, *Schisandra sphenanthera* has also been reported to have anti-aging effects, potentially restoring immune function by counteracting immunosenescence, the gradual decline of the immune system with age [96].

Tetrandrine, the active component of *Stephania tetrandra* (fangji), has shown potential in reducing glomerulosclerosis in rats [89], and is suggested for use in combination with glucocorticoids, to maximize therapeutic efficacy. The application of *Stephania tetrandra* may also mitigate the adverse effects associated with glucocorticoid therapy [97]. Combination therapy could be beneficial in idiopathic nephrotic syndrome, where the exact pathogenesis may vary. Such a combination might balance therapeutic effects while reducing potential adverse impacts. Fangji, when used with Polyporaceae (fuling), has been reported to improve renal function; however, the specific role of tetrandrine as an active component remains underexplored in these studies [98]. Additionally, fangji combined with Huangqi (Astragali Radix) was found effective for treating nephrotic syndrome within an integrative-systems pharmacology approach [99]. Knowledge regarding the nephrotoxicity of fangji remains limited, and treatment duration and dosage are critical factors influencing its toxicity [97]. Further studies are needed to elucidate its clinical role in treating proteinuria in kidney patients. Both tetrandrine and Huangqi have been identified as potent cellular autophagy activators against reactive oxygen species [100,101]. However, there is still a lack of evidence regarding the effects of tetrandrine or Huangqi on cellular senescence

Ganoderma extracts exhibit antioxidant properties that defend against podocyte damage from oxidative stress and apoptosis [102]. Ganoderma lucidum has a long history in Chinese medicinal use, dating back to 100 AD [102], of promoting health and longevity [103]. Cochlearol A, a meroterpenoid isolated from the Ganoderma genus—specifically, Ganoderma cochlear—has demonstrated renoprotective effects, and can be successfully synthesized [104]. *Silybum marianum* has also been identified as a senolytic agent targeting senescent cells [105], with reported effects in reducing levels of senescence-associated beta-galactosidase (SA-β-Gal), p16INK4A, and SASP markers [106]. Additionally, Ganoderma lucidum, Ganoderma meroterpenoids, and Ganoderma triterpenes have each been found to target different intracellular signaling pathways to achieve renoprotective activity, particularly in counteracting cisplatin side effects [107]. In addressing oxidative stress as a cause of kidney damage, *Silybum marianum* is recognized as a potent antioxidant with the potential to protect the kidneys from reactive oxygen species, although further studies are needed to elucidate its mechanism [108]. Research has shown that the antioxidant and anti-apoptotic properties of silymarin exert protective effects against kidney damage induced by vancomycin-related oxidative stress and nephrotoxicity [109,110]. Recent reviews of Ganoderma lucidum highlight its anti-aging, anti-senescence, anti-cancer, and antioxidant properties, suggesting its potential as a therapeutic strategy for various age-related diseases [103]. Furthermore, Ganoderma lucidum has been associated with key signaling pathways related to healthy aging, such as Nrf2, mTOR, and MAPK pathways [111].

*Plantago asiatica* shows promise as a potential treatment for nephrotic syndrome due to its multifunctional properties, including anti-inflammatory, anti-apoptotic, and renoprotective effects observed in nephrotic syndrome rat models [112]. Hispidulin, a flavonoid component isolated from *Plantago asiatica* [113], has been reported to reduce podocyte damage induced by high-glucose conditions by regulating autophagy and the Pim1-p21-mTOR signaling axis [114,115]. Hispidulin is considered a potential lead compound for further development and investigation in treating diabetic nephropathy [115].

Rutin is a flavonoid with antioxidant properties that can reduce oxidative stress, thereby attenuating gentamicin-induced nephrotoxicity [116]. However, the protective role of rutin in the autophagy of kidney cells remains unknown [116]. Pinzaru et al. reported that rutin can induce senescence activities, while also exhibiting anti-cancer effects through various signaling pathways, including MAPK, PI3K/Akt, Wnt/β-catenin, and p53. Although rutin has shown beneficial effects, our understanding of its impact on kidney health is still limited, and further studies and clinical research are needed

Another lesser-known compound is leonurine. Dexamethasone, a glucocorticoid commonly used in CKD patients, has been associated with cytotoxicity. Recent findings indicate that leonurine may have protective effects against dexamethasone-induced cytotoxicity, though further studies are required to confirm and validate this finding [117]. The antioxidant role of leonurine in podocyte injury remains unclear [57], and additional research on leonurine in kidney-related contexts is warranted. Chen et al. and other researchers reported that leonurine could delay the senescence process via activation of the Nrf2 signaling pathway [118,119].

Resveratrol has demonstrated the ability to activate SIRT1-stimulated autophagy, which can enhance kidney cell adaptation to hypoxia and prevent apoptosis in rats [120]. Besides resveratrol, astragaloside IV [121], curcumin [122], hispidulin [115], cordyceps [123], poria cocos wolf [124], and Tripterygium glycoside [125] also displayed a remarkable ability for the activation of autophagy. Our understanding of autophagy remains limited, particularly in the context of human glomerular diseases [28]. Up to this date, there is a lack of studies on hispidulin’s role in CS, but it was reported to have an active role in autophagy by activating AMPK and inhibiting the mTOR signaling pathway [126]. Unfortunately, there was a lack of direct evidence regarding hispidulin and cellular senescence, but it was believed that they are closely associated with each other, given their involvement in autophagy. Nevertheless, the abovementioned natural products provided insight into autophagy as a potential treatment direction.

Studies on the natural products listed in Table 1 have been limited outside of China, necessitating further investigation to better understand their potential in treating kidney diseases. Many studies on these natural products have been conducted in China, and some may be categorized as Traditional Chinese Medicine (TCM). Exploring the relationship between these products and key regulators of aging, such as sirtuin, mTOR, and AMPK, could help assess their potential efficacy in kidney disease treatment, and potentially offer new therapeutic options. It is important to note that while some natural products may be used within the TCM framework, not all natural products fall under this system of medicine.

The metabolites and signaling pathways in Figure 2 offer a comprehensive view of cellular senescence and autophagy, as these pathways are extensive and vary depending on cell type, duration of stressors, and type of stimuli. Figure 2 highlights the relevance of natural products used in CKD management with respect to cellular senescence and autophagy, providing insights into their potential mechanisms for ameliorating CKD. Figure 2 illustrates the fact that under normal, stimuli-free conditions, GATA4 is degraded and regulated by SQSTM1/p62, resulting in a transient presence of SASP and effective immune clearance of senescent cells. In experimental rat models, p21 was found to be transiently elevated, whereas p16 levels increased during sustained cellular senescence [15]. Sirtuin, mTOR, and AMPK in Figure 2 are key regulators of cellular senescence, aging, and various age-related diseases. The role of sirtuins in aging and age-related diseases has been well established in recent studies. The major molecular pathways involved in cellular senescence appeared to be p53, p21, p16, and NF-kb, and sirtuin was reportedly having a direct impact on these molecules [15,127,128]. Figure 2 illustrates the fact that p21 transient presence helps with immune cell clearance, hence inhibiting aging, and the DNA damage prevents the p21 from clearing the immune cells. Cellular senescence can occur in both a programmed and a stress-induced manner, involving distinct pathways and processes [34]. Senescence can develop independently through p21 without involving DNA damage response (DDR) and p53, whereas acute and chronic senescence are more complex, engaging various stressors such as p16, DDR activation, p53, and p21 [34]. Consequently, programmed senescence may not lead to cellular senescence accumulation; instead, senescence-related stressors may accumulate as a collateral, detrimental effect. When a certain level of DNA damage is detected, the release of the senescence-associated secretory phenotype (SASP) follows; if SASP persists, it can lead to cellular senescence, which later contributes to CKD. The green arrow in Figure 2 indicates the role of GSK3, an enzyme that has two highly conserved isoforms, GSK3α and GSK3β. GSK3β has been reported to directly impact the Wnt signaling pathway, autophagy, and podocyte senescence [35]. Studies suggest that GSK3β acts as an inhibitor of the Wnt signaling pathway [35], while conversely, the Wnt signaling pathway can inhibit GSK3β [129]. The Wnt pathway activates mTOR by inhibiting GSK3β; thus, autophagy activation may positively influence cellular senescence. This bidirectional regulation between the Wnt signaling pathway and GSK3β has differential effects on autophagy and cellular senescence, requiring further studies to clarify the contradictory role of GSK3. The authors believe that exploring the specific underlying conditions related to GSK3 would further enhance our understanding of its role in autophagy.

## 6. Advancing CKD Treatment: Insights into Natural Products, TCM, and Personalized Therapies

Table 1 presents an overview of natural products employed in the management of chronic kidney disease (CKD) patients, while Table 2 focuses on studies exploring the involvement of autophagy in kidney diseases. Importantly, research into the autophagy-related effects of these natural products remains predominantly at the preclinical stage, and is primarily conducted in animal models, as detailed in Table 2. Additionally, Table 1 outlines the potential limitations of each natural product, including adverse effects, toxicity, and poor bioavailability. Investigating natural products from an autophagy perspective is crucial for advancing therapeutic strategies and enhancing our understanding of the underlying pathogenesis of CKD. Furthermore, preclinical studies of natural products in autophagy should be expanded to address kidney aging. While autophagy research in CKD is essential to uncover the therapeutic characteristics of naturally derived compounds, focusing on kidney aging could offer a distinct perspective by emphasizing pathways such as sirtuin, mTOR, and AMPK regulators. This approach may provide novel insights into the intersection of autophagy and kidney aging, advancing the development of targeted therapies and the pathogenesis of the diseases. Despite the promising potential of natural compounds in treating chronic kidney disease (CKD), there is a notable scarcity of randomized clinical trials (RCTs) evaluating their efficacy, particularly concerning autophagy-focused therapies. This lack of rigorous clinical evidence hinders the integration of these compounds into current therapeutic strategies for CKD. To establish their safety and effectiveness, comprehensive RCTs are essential. Such studies would provide the necessary data to determine the viability of incorporating natural products into standard CKD treatments, especially those targeting autophagy pathways.

Well-designed clinical studies on these renoprotective natural substances could shed light on kidney diseases that are currently not fully understood. The “one drug, one target” approach of mainstream clinical medication [130] often focuses on symptom relief, aiming for rapid alleviation. However, the disappearance of symptoms does not equate to addressing the underlying causes of disease, leading many patients to experience multiple relapses. In contrast, Traditional Chinese Medicine (TCM) generally exerts therapeutic effects gradually and holistically, treating the human body as an interconnected system [76,130]. This perspective suggests a significant leap for researchers, as conventional pharmacological methods are often inadequate for elucidating the therapeutic mechanisms of TCM. By exploring what can be derived and scientifically proven from TCM and other natural products, researchers may gain valuable insights into the pathogenesis and mechanisms of poorly understood kidney diseases. It is also crucial to deepen our understanding of these herbs, as even nontoxic herbs can cause adverse effects if used inappropriately (Wang et al., 2018). The therapeutic effects of TCM result from the synergistic action of multiple pharmacologically active compounds [130]. The relationship between natural products used in Traditional Chinese Medicine (TCM) and their effects on sirtuin, mTOR, and AMPK pathways requires further study. For instance, curcumin and resveratrol have been well-studied for their effective roles in activating sirtuin, inhibiting mTOR, and activating AMPK to promote autophagy; however, similar studies on other natural products are still lacking.

It is notable that nearly all natural products reported to have renoprotective effects primarily possess anti-inflammatory and antioxidant properties. Renal oxidative stress plays a critical role in cellular senescence [13]. These two characteristics may enable natural products to act as protective barriers against free radicals and modulate Treg cell activity. Additionally, some natural products listed in Table 1 may alleviate the adverse effects of common clinical medications, such as cyclophosphamide, tacrolimus, and dexamethasone, or offer complementary effects when used alongside tetrandrine with glucocorticoids. Natural products can be an appropriate fit in personalized treatment plans, as they may complement mainstream medications and help mitigate their side effects. Biomarker levels can vary significantly among individuals with the same disease diagnosis, indicating the need for tailored treatment approaches. Although individually tailored treatment plans have not yet been widely reported, implementing personalized treatment requires a deep understanding of each patient’s unique medical situation. This challenge may gradually be addressed by introducing herbs or natural products as supplementary therapies. To maximize therapeutic effects for CKD patients, a comprehensive understanding of the characteristics of natural products—particularly in relation to autophagy and cellular senescence (CS)—is needed through in-depth clinical studies and preclinical studies. For instance, the poor bioavailability and absorption of curcumin have posed significant challenges in fully harnessing its therapeutic potential. However, the use of piperine has been shown to significantly enhance the bioavailability of curcumin [131]. Further studies of this nature are essential to deepen our understanding of naturally derived compounds. Addressing the complexities and unanswered questions surrounding these compounds will bring us closer to their effective application in therapeutic strategies.

Prescribing natural products as supplements alongside standard clinical medications has the potential to enhance treatment outcomes. However, extensive research is still required to substantiate these complementary practices, as many compounds listed in Table 1 have been reported to exhibit toxicity and side effects, including nausea, vomiting, diarrhea, nephrotoxicity, and poor absorption. While certain elements of Traditional Chinese Medicine (TCM) may exhibit toxic properties at specific levels, these characteristics can sometimes stimulate beneficial immune responses that support recovery [132]. However, the effective and safe use of TCM necessitates rigorous standardization, thorough scientific examination, precise dosage measurement, and systematic observation and reporting. Significant gaps remain in the evidence base for these practices, particularly regarding their safety, efficacy, and mechanisms of action. Moreover, due to the experiential nature of TCM—heavily reliant on the expertise and judgment of practitioners—alongside the minor toxic properties used to stimulate immune responses and the variability in individual responses, large-scale clinical trials are essential. Such studies could provide detailed insights into achieving optimal dosage levels while addressing critical issues such as safety and bioavailability. By bridging these gaps, future research could improve the integration of TCM into modern therapeutic strategies, enhancing its safety and efficacy for diverse populations.

**Table 1 ijms-26-00003-t001:** Renoprotective herbs/natural products.

Name (with Active Component)	Common Name	The Common Usage in CKD	Description/ Usefulness	Drawback/ Limitation	Targeting Molecule/ Signaling Pathway
Turmeric (Curcumin)	-	-diabetic nephropathy	-demonstrated promising renoprotective [49,122,133] ability against nephrotoxicity [73,134] and renal injury [135] -reduces proteinuria in lupus nephritis [136,137] -enhances autophagy of podocytes [122] -significant antioxidant protective effect on renal ischemia-reperfusion injury [138]	-poor adsorption, which affected its efficacy [122] but is not a significant concern [139]	-Transforming growth factor B and interleukin-8 [140] -NF-KB and activation of the JAK2/STAT3 signaling pathway [141] -Nrf2 [49]
Resveratrol (polyphenolic)		-diabetic nephropathy	-renoprotection from Adriamycin induced-FSGS [142] -ameliorates podocyte damage [143], the potential treatment approach for diabetic nephropathy patients [144] -Resveratrol’s renoprotective effects work by an activated mechanism to inhibit oxidative stress [145] and apoptosis of mitochondria [143] and podocyte [144,146]	-poor solubility and limited bioavailability [147] -molecular mechanism [144], pharmacokinetic, and pharmacodynamics need further studies [148]	-C3aR and C5aR [142] -Nrf2 activation [145] -sirtuin 1 (sirt 1) activation [145,149] -angiotensin type 1 receptor and NF-κB [149] -downregulates malondialdehyde and inhibit reactive oxygen species (ROS) [143] -5′ adenosine monophosphate-activated protein kinase (AMPK) [144]
*Astragalus membranaceus* or *Astragalus mongholicus* (Astragaloside IV)	Huáng Qí	-effective adjuvant therapy used in membranous nephropathy [61] -alternative therapy for the frequent-relapse nephrotic syndrome [66]	-attenuation of podocyte injury [61] -most essential herbs to treat proteinuria [66] -ability to restore actin cytoskeleton in podocytes [61] -reduce chemotherapy toxicity [150], such as cyclophosphamide-induce toxicity [66,151,152]	-side effects are not well understood because they are generally used in combination with other herbs [5] -lack of molecular studies	-reduction in phosphorylation of JNK and ERK and their signaling pathway [61]
*Coptis chinensis* (Berberine)	Huang Lian/ Coptis rhizome	-commonly used to treat diabetes	-berberine exhibits renoprotective effects against various podocyte injury agents [153,154,155,156,157,158]	-lack of studies	-inhibition of RhoA/ROCK signaling was reported -EP4-Gαs-cAMP signaling pathway as significant renoprotective effects of berberine [82] -NF-κB signaling pathway [156] -β-arrestins, intercellular cell adhesion molecule-1 (ICAM-1), and vascular cell adhesion molecule-1 (VCAM-1) [158] -reduces PKC- β [155]
*Rheum palmatum* L. (emodin and rhein)	Rhubarb/Dahuang	-treating diabetic kidney disease [82]	-improvement of renal function and reduction of proteinuria was reported [5] -Rhein could prevent kidney damage [82] -emodin was reported to have a protective action on podocytes [159]	-nausea, vomiting, diarrhea, and abdominal pain were reported [5]	-downregulate tumor necrosis factor-α and interleukin-6 [160] -Rhein was reported as downregulating the wnt/ ß-catenin signaling pathway and upregulating SIRTI [82] -Emodin was reported to reduce glycation of proteins, inhibit cFLIP, TGF-ß1, and p38MAPK pathways [82] -Emodin inhibits the PERK-eIF2α signaling pathway [159]
*Tripterygium wilfordii* Hook F (TWHF) Triptolide (PG-90) (diterpene triepoxide)	Lei Gong Teng	-used to treat glomerulonephritis for more than 30 years in China [5]	-outstanding antiproteinuric effects are proven in animal models [5] -reduces podocyte injury via podocyte apoptosis inhibition [161,162] -improved kidney function compared with cyclophosphamide was reported [163]	-lack of clinical data in English -nephrotoxicity as a side effect was reported due to oxidative stress [164] -insufficient evidence to prove as effective as prednisone and cyclophosphamide [163]	-nuclear factor-kappa B (NF-κB) [165,166] -inhibit C5b-9-induced MAPK activation [78] -Triptolide was reported to decrease chemokine expression and inhibit macrophage infiltration [82] -antioxidative to inhibit reactive oxygen species [82]
*Ganoderma fungus (lucidum/cochlear)* (Triterpenes)	Lingzhi	-various acute kidney diseases and chronic kidney diseases [107]	-Ganoderma lucidum has low toxicity with significant antioxidant activity [102,167] -a successful clinical outcome of suppressing proteinuria in FSGS patients [168] -enhance antioxidant ability with vitamin C and E that successfully suppress proteinuria [169]	-suggested using with caution as potential toxicity was reported [170]	-CD4^+^, CD25^+^, Foxp3^+^, Treg, Interleukin-10, Breg [171] -interleukin-8 [102]
*Cordyceps sinensis* and *Cordyceps militaris* (cordycepin)	Dong Chong Cao	-commonly used in diabetes nephropathy	-attenuates glomerular damage when used together with TWHF [82,123] -antinephritic function [172,173] and significant nephroprotective effect [87] -cordycepin has excellent effect on anti-hyperuricemic [174]	-molecular mechanism unclear [123]	-podocin, nephrin [82] -Toll-like receptor 4/ nuclear factor kappa B [123,175]
*Plantago asiatica* and *Plantago major* (Hispidulin)		-diabetes nephropathy	-prevents podocyte apoptosis via autophagy [115] -renoprotective effect and potential therapeutic effects in nephrotic syndrome [112] -*Plantago major* could improve kidney function and reduce apoptosis [112] -improved proteinuria [112,176]	-unclear exact mechanism -renoprotective effect in nephrotic syndrome was reported in animal models only	-Mitogen-activated protein (MAP) [112]
*Schisandra sphenanthera* (deoxyschizandrin)	Wuzhi	-commonly reported to be used after renal transplant	-enhanced tacrolimus effect when used in combination [177,178,179,180], while it did not increase the adverse effect [66] -nephrotoxicity protective effect induced by cisplatin [181]	-limited evidence and research on human clinical studies [180].	-transcription factor NF-E2-related factor 2 [181]
*Stephania tetrandra* S Moore (Tetrandrine)	Fangji	-nephrotic syndrome	-tetrandrine is a practical component in the treatment of nephrotic syndrome [182] -effective combined treatment with glucocorticoids (prednisolone) could be another beneficial approach for the nephrotic syndrome [99,183]	-identification of therapeutic molecule was narrowed down but is still unclear and pending further validation [184] -potential nephrotoxicity [97]	-inhibits RhoA activation [185]
Polyphenol (Epigallocatechin-3-gallate (EGCG))	Green tea	-diabetic nephropathy	-powerful antioxidant against oxidative stress [58,186,187] -promising therapeutic effects on various kidney diseases [8]	-unstable bioavailability is the concern [8] -lack of clinical studies on humans to provide conclusive evidence	-nuclear factor erythroid related-factor 2 (Nrf2) [8,60,188] -nuclear factor κB [8,60]
*Silybum marianum* (silymarin)	milk thistle	-protects against diabetic nephropathy [109]	-promising nephroprotective [108,109,110,189] ability against a wide range of drugs such as cyclosporin [90] -reduces urinary excretion of albumin [190]	-first reported case on the usage of Silybum that was not effective [189,191] -challenging bioavailability to show natural effect [192]	-blocking TNF-induced activation of NF-κB [109]
*Herba Leonuri* Leonurine	Motherwort	-renal fibrosis	-kidney protective effects on animal models [57] -a potential therapeutic drug to prevent podocyte injury from oxidative stress with its antioxidant ability [57]	-lack of clinical studies [193] -minimal scientific studies on kidney	-TGF-β/Smad3 and NF-κB signaling pathways [193]
Rutin		-type 1 diabetic mice [194]	-anti-inflammatory effect and anti-oxidative-stress effect on the kidney [195,196] -improved kidney structure [197]	-lack of studies on kidney-related disease and clinical studies	-TGF-β1/Smad3 signaling [198] -ceramide, MAPK, p53, and calpain [195]
Polyporaceae *Poria cocos* Wolf	Fu-Ling	-commonly used to treat renal fibrosis	-strong ability to inhibit renal fibrosis and podocyte injury [199]		-suppressed TGF-β/Smad pathway [200]

**Table 2 ijms-26-00003-t002:** Natural products and their active ingredients: studies of autophagy in kidney diseases.

Natural Ingredient	References				Disease/ Target	Conclusion
Curcumin	Animal studies in kidney autophagy:	Animal Model	Dosage and route of administration	Combination		
	[201]	Thirty male Sprague Dawley rats	300 mg/kg/d, (Intragastric)., for 8 weeks	Treatment with curcumin only	DN	Curcumin treatment protects DN by inducing autophagy and alleviating podocyte EMT, through the PI3k/Akt/mTOR pathway.
	[202]	Male Wistar rats, Sprague Dawley (SD) rats, and New Zealand white rabbits	300 mg/kg/d, (Intragastric)., for 30 days	Treatment with curcumin only	Membranous Nephropathy (MN)	Curcumin enhances kidney health by promoting autophagy and reducing oxidative stress in the kidneys through specific pathways like PI3K/AKT/mTOR and Nrf2/HO-1.
	[122]	male balb/c mice	200 mg/kg/d, (intragastric) for 8 weeks	Treatment with curcumin only	Diabetic Nephropathy (DN)	Curcumin inhibited podocyte apoptosis and accelerated cell autophagy via regulating Beclin1/UVRAG/Bcl2, demonstrating that curcumin exerted significantly protective effects in DN.
	[203]	Twenty-four Sprague Dawley male rats, unilateral ureteral obstruction (UUO) rats	200 mg/kg (gastrogavage) for 14 days	Treatment with curcumin only	Renal interstitial fibrosis (RIF)	Promising treatment for RIF and its antifibrotic effect may be regulated by autophagy and protection of mitochondria function.
	[204]	Nude Mice	100, 200, and 400 mg/kg via oral gavage for 30 days	Treatment with various concentration of curcumin	Renal cell carcinoma	Curcumin is capable of inhibiting the proliferation of renal cell carcinoma by regulating the miR-148/ ADAMTS18 axis via the suppression of autophagy in vitro and in vivo.
	Clinical studies with kidney autophagy:					
	No record found					
	Clinical studies with kidney patients:	Type of clinical trial	Dosage and route of administration	Combination		
	(2011–2017): [48,49,140,205,206]				Chronic kidney disease (CKD)/Diabetic kidney disease (DKD)/ Hemodialysis (HD)	Short-term turmeric supplementation could attenuate proteinuria and has potential to reduce oxidative stress. Lack of clinical studies to verify appropriate dose for long-term safety usage in all stages of CKD. Short-term use of curcumin halts the progression of diabetic kidney disease. Curcumin considered as an effective anti-inflammatory supplement in HD patient group.
	[137]	randomized, double-blind clinical trial n = 46	Dosage: 500 mg of curcumin per capsule, taken three times daily after meals (total 1500 mg/day) for 16 weeks.	Adjuvant therapy	DN	Curcumin could be an effective adjuvant therapy for ameliorating proteinuria in type 2 diabetes patients.
	[207]	double-blind randomized pilot study n = 31	Taken orally in100 mL of orange juice containing 12 g of carrot and 2.5 g of turmeric, given after each hemodialysis session (three times per week) for 3 months	nutritional supplement (curcumin-enriched juice)	HD	Short-term treatment with curcumin suggesting that oral supplementation of curcumin may have anti-inflammatory effect in HD patient group.
	[208]	single-center, prospective, double-blinded, randomized controlled trial n = 60	Curcuminoids were administered orally at a dosage of 1500 mg per day, starting 3 days before and continuing 2 days after the coronary procedure for 5 days (3 days prior to and 2 days post procedure).	Curcuminoids were used as an anti-inflammatory and antioxidant supplement in addition to the standard prophylaxis protocol for CI-AKI.	Contrast-induced acute kidney injury (CI-AKI)	Reduces the overall CI-AKI and AKI incidents in CKD patient undergoing elective coronary procedure.
	[209]	randomized, double-blind, placebo-controlled trial. n = 43	Curcumin was administered orally at a dosage of 1 g/day for 12 weeks.	Curcumin was used as a nutritional supplement; the placebo group received corn starch.	HD	Potential effects on antioxidant response, but insufficient to reduce oxidative stress markers and inflammation in hemodialysis patients
	In vitro studies:					
	[210]				Renal fibrosis	Curcumin-induced autophagy extracellular vesicles improved fibrosis condition.
Curcumin’s role in autophagy in kidney disease was investigated in animal studies, and there was no record of clinical studies in term of autophagy in kidney disease found so far, to the best of the authors’ knowledge, from 2019 to 2023. There are clinical studies of curcumin in the cancer patient group, and in cardiovascular disease and other diseases such as Alzheimer, but not in kidney autophagy [211].						
Astragaloside IV (AS-IV)	Animal studies in kidney disease	Animal model	Dosage and route of administration	Combination		
	[212]	50 Wistar rats: 33 rats were confirmed with adriamycin nephropathy	Astragaloside: intragastric 150 mg/kg/day for 3 months. Methylprednisolone (MP): oral gavage at a dosage of 20 mg/kg/day for 3 months.	Adjuvant therapy, Methylprednisolone (MP): used as a reference standard therapy for nephrotic syndrome	Adriamycin nephropathy	AS-IV could prevent the progression of kidney injury.
	[213]	48 diabetic male Sprague Dawley rats.	Astragaloside IV: administered via intragastric at doses of 2.5 mg/kg/day, 5 mg/kg/day, and 10 mg/kg/day for 12 weeks. Tempol: administered via drinking water at a concentration of 1 mmol/L for 12 weeks. Insulin: Administered via subcutaneous injection at 6 U/day for 12 weeks.	Adjuvant therapy with tempol: used as a reference antioxidant therapy. Insulin: served as baseline treatment for diabetic rats.	DN	AS-IV could prevent kidney injury in DN rat model.
	[214]	18 male and female Sprague Dawley rats	Astragaloside IV (ASI): administered via intragastric (i.g.) administration at a dosage of 40 mg/kg/day for 8 weeks.	Adjuvant therapy	DN	AS-IV exerts therapeutic effect on DN, potentially through the inhibition of excessive mesangial proliferation and renal fibrosis via the TGF-β1/Smad/miR-192 signaling pathway.
	[215]	32 male Sprague Dawley rats	AS-IV was administered via oral gavage at doses of 20, 40, and 80 mg/kg once daily. Positive control rats received Metformin 200 mg/kg via oral gavage for 8 weeks.	Treatment with astragaloside IV only. Metformin as positive control.	DN	AS-IV could exert protective effect from ER stress-induced apoptosis via the downregulation of p-PERK, ATF4 and CHOP.
	[216]	32 male Wistar rats	AS-IV was administered via intragastric (i.g.) gavage at a dose of 40 mg/kg/day for rats for 11 days.	Treatment with astragaloside IV only.	podocytes	AS-IV could alleviate PAN-induced podocytes injury via partial activation of Wnt/planar cell polarity (PCP) pathway.
	[217]	8-week-old male db/db mice	AS-IV: 5 g/kg in the diet. Enalapril: 0.8 g/kg in the diet.Both compounds were administered as dietary supplements. Combination therapy included both AS-IV and Enalapril at the same doses for 12 weeks.	AS-IV was used alone and in combination with Enalapril (an ACE inhibitor).	DN	AS-IV is more effective when used in combination with angiotensin, converting enzyme inhibitor to exert renal protective effect.
	[218]	16 male Sprague Dawley (SD) rats	Astragaloside IV (AS-IV) dissolved in 0.5% CMC-Na, 80 mg/kg/day administered via oral gavage for 12 weeks.	Treatment with astragaloside IV only.	DN	AS-IV treatment could inhibit inflammation in rats’ kidney; hence, it ameliorated the severity of DN.
	[219]	40 male Sprague Dawley (SD) rats	40 mg/kg and 80 mg/kg of AS-IV administered daily via intragastric for 10 weeks.	Treatment with astragaloside IV only.	DN	AS-IV can ameliorate renal injury caused by high glucose; it has anti-oxidative-stress, anti-inflammatory, and anti-epithelial-mesenchymal transition (EMT) effects, and can inhibit the Wnt/β-catenin signaling pathway.
	[69]	Male diabetic C57BLKS/J-LepR (db/db) mice	2 mg/kg/day, 6 mg/kg/day, and 18 mg/kg/day of AS-IV administered via oral gavage for 8 weeks.	Treatment with astragaloside IV only.	DN	AS-IV protects podocytes from apoptosis by inhibiting oxidative stress via activating PPARγ-Klotho-FoxO1 signaling pathway.
	[220]	24 Male Sprague Dawley (SD) rats	ADR (adriamycin) was administered intraperitoneally at 4 mg/kg in four equal injections over 5 weeks. AS-IV was administered intragastrically at 10 mg/kg daily for 5 weeks.	AS-IV was combined with ADR treatment	Adriamycin-induced renal damage	ASIV might protect nephrocytes against ADR-induced ferroptosis, potentially via the activations of the Pi3K/Akt and Nrf2 signaling pathways.
	[65]	40 male Sprague Dawley (SD) rats	40 mg/(kg·d) of AS-IV administered via intragastric infusion for 8 weeks.	No combination. Astragaloside IV and SRT1720 were tested separately.	Kidney aging	AS-IV can delay kidney aging by regulating the SIRT1/p53 signaling pathway.
	[221]	50 db/db mice	AS-IV at 10 mg/kg/day (low dose), 20 mg/kg/day (medium dose), and 40 mg/kg/day (high dose) via oral gavage for 12 weeks.	Treatment with astragaloside IV only.	EMT/DKD	C-X3-C motif ligand 1 (CX3CL1) plays a significant role in the progression of EMT; it is a potential target for AS-IV to alleviate renal tubular EMT.
	[68]	32 8-week-old male Sprague Dawley rats	40 mg/kg/day and 80 mg/kg/day of AS-IV administered via oral gavage for 12 weeks.	Treatment with astragaloside IV only.	DN	AS-IV upregulate the klotho expression, hence exert a protective effect.
	[222]	40 Male Sprague Dawley rats	Astragaloside IV (AS) dose: 40 mg/kg/day via oral gavage for 8 weeks.	No combination therapy mentioned; AS and UBCS039 were administered as monotherapies in separate groups.	DKD	AS-IV inhibited podocytes pyroptosis in DKD by regulating SIRT6/HIF-1α pathway, thus, ameliorating injury of DKD.
	Animal studies in kidney autophagy					
	[223]	6-week-old male C57BL/6J mice	Astragaloside IV (AS-IV) doses: 3 mg/kg, 6 mg/kg, and 12 mg/kg via oral gavage for 8 weeks, once daily	Treatment with astragaloside IV only.	DN	AS-IV was suggested to prevent the progression of DN by SERCA2-dependent ER stress attenuation and AMPKα-promoted autophagy induction.
	[224]	KK-Ay mice: used as a model for DKD with renal lesions resembling those in human type 2 diabetes mellitus. C57BL/6J mice: used as normal controls.	40 mg/kg/day of AS-IV via oral gavage for 12 weeks	Treatment with astragaloside IV only.	EMT	AS-IV could exert protective effect on podocyte from EMT via the modulation of SIRT1–NF-κB pathway and autophagy activation.
	[225]	40 Male Sprague Dawley (SD) rats,	Cisplatin: 15 mg/kg, intraperitoneally. AS-IV (oral gavage) Low dose: 40 mg/kg. High dose: 80 mg/kg. AS-IV: administered daily for 7 days. Cisplatin: administered on the 4th day of the 7-day AS-IV treatment.	Cisplatin was combined with AS-IV in the treatment groups	Cisplatin-induced liver and kidney injury	AS-IV induced autophagy and limit the expression of NLRP3 to effectively protect against cisplatin-induced injuries.
	[121]	30 Male Sprague Dawley (SD) rats,	Astragaloside IV (AS-IV): Low dose: 10 mg/kg/day. High dose: 20 mg/kg/day via oral gavage for weeks. Benazepril: 1 mg/kg/day via oral gavage for 7 weeks.	Separate groups treated with the following: low dose of AS-IV (10 mg/kg/day). high dose of AS-IV (20 mg/kg/day). Benazepril (1 mg/kg/day) for comparison.	chronic glomerular nephritis (CGN)	AS-IV improved kidney function, reduced kidney lesion, and was remarkable in inhibiting the activation of PI3K/AKT/AS160 pathway and improving autophagy activation.
	[226]	36 Male Sprague Dawley (SD) rats,	Astragaloside IV (AS-IV): 80 mg/kg/day via oral gavage for 8 weeks. Metformin (Met): 200 mg/kg/day via oral gavage for 8 weeks.	AS-IV and Metformin were administered as separate treatment groups for comparison	Type 2 diabetes liver injury	AS-IV alleviated diabetic liver injury in T2DM rats, and it could promote AMPK/mTOR-mediated autophagy.
	[227]	40 Male Sprague Dawley (SD) rats,	ASIV doses: 40 mg/kg for the ASIV-40 group. 80 mg/kg for the ASIV-80 group. Administration route: oral gavage with ASIV dissolved in 0.5% CMC-Na solution for 4 weeks.	Treatment with astragaloside IV only.	CKD	Astragaloside IV exerts an anti-fibrosis effect and could enhance ALDH2 transcriptional activity. ALDH2-mediated autophagy could be a novel target for treating renal fibrosis.
	Clinical studies of autophagy in kidney patients					
	No records					
	Clinical trial studies in kidney patients	Type of studies	Dosage and route of administration	Combination		
	[228]	Pragmatic, assessor-blind, parallel, randomized controlled clinical trial n = 118	30 g/day astragalus daily (oral administration) for 48 weeks.	Adjuvant therapy with antidiabetic agents (e.g., insulin) and renin-angiotensin system (RAS) blockers like ACE inhibitors	DKD	This trial evaluates astragalus’s effectiveness in slowing DKD progression and identifies predictors for personalized use. Preliminary results are promising, with objective outcomes minimizing bias and supporting integration of conventional and Chinese medicine.
	In vitro studies					
	[229]				EMT	AS-IV blocked the mTORC1/p70S6K signaling pathway in renal tubular epithelial cells, hence, ameliorating high glucose-mediated renal tubular EMT.
	[230]				Glomerular Endothelial Cells	AS-IV can maintain the integrity of the filtration barrier in glomerular endothelial cells under diabetic conditions.
	[231]				DKD	AS-IV improved mitochondria function and protected podocytes from apoptosis and resistance to oxidative stress-induced diabetic kidney injury. The process was believed to be closely associated with the activation of Nrf2-ARE/TFAM signaling.
AS-IV clinical studies are lacking but animal studies and in vitro studies suggested that AS-IV could exert renoprotective effect and activate autophagy.						
Epigallocatechin-3-gallate (EGCG)	Animal studies in kidney disease	Animal model	Dosage and route of administration	Combination		
	[232]	Male Sprague Dawley (SD) rats	Indomethacin: 10 mg/kg. L-NAME: 10 mg/kg. Iopromide: 1.8 g(I)/kg. EGCG (Epigallocatechin gallate): 5, 10, or 20 mg/kg. ZnPP (HO-1 inhibitor): 30 mg/kg. Administration route: Indomethacin, L-NAME, and iopromide: intravenous (via left external jugular vein). EGCG: intravenous. ZnPP: intraperitoneal injection.	The CIN model included sequential injections of indomethacin, L-NAME, and iopromide. EGCG was tested alone for its protective effects. ZnPP was used as an inhibitor to assess the role of HO-1 in the protective mechanisms.	Contrast-induced nephropathy (CIN)	EGCG is a potent inducer of the antioxidant (heme oxygenase-1) that can protect CIN by ameliorating oxidative stress and inflammation.
	[233]	20 male Wistar rats	EGCG: 50 mg/kg/day was administered via intraperitoneal injection. Duration: EGCG treatment lasted for 9 days. Started 2 days before UUO surgery. Continued during the 72 h obstruction period. Maintained for 5 days after UUO reversal.	Treatment with EGCG only.	Ureteral obstruction (UO)	EGCG was reported to have no significant protective effect on glomerular function when measured post reversal of unilateral ureteral obstruction but attenuated some of the kidney injury markers and pro-inflammatory mediators.
	[234]	24 Dahl salt-sensitive (Dahl/SS) rats,	Epigallocatechin-3-gallate (EGCG): 50 mg/kg body weight via oral gavage twice daily for 6 weeks.	EGCG was the sole treatment in the EGCG group; no other medications were combined.	Renal damage in Dahl rats with salt hypertension	EGCG may exert antioxidant, anti-inflammatory and apoptosis-inducing effect on fibroblasts to attenuate renal damage and salt-sensitive hypertension.
	[235]	30 male Wistar rats	EGCG at 50 mg/kg and 100 mg/kg, administered daily (oral gavage) for 90 days.	EGCG was the sole treatment tested.	Cigarette smoke induced renal and hepatic fibrosis	EGCG ameliorates renal and hepatic oxidative stress and inflammation, and could attenuate renal and hepatic fibrosis.
	[59]	32 Male Sprague Dawley rats	40 mg/kg/day EGCG for the low-dose group. 80 mg/kg/day EGCG for the high-dose group. Duration: 8 weeks, both high and low dose administered via oral gavage.	EGCG was the sole treatment tested.	Type 2 diabetic rats	EGCGwas reported to have renoprotective effects on type 2 diabetic rats mainly via repression of endoplasmic reticulum stress-mediated NLRP3 inflammasome.
	In vitro studies					
	[236]				Epithelial mesenchymal transition (EMT)	EGCG could prevent the epithelial mesenchymal transition of the renal tubular cells via nrf2 pathway.
	[237]				Reactive oxygen species	EGCG can improve the antioxidant capacity of the cell to promote repair of the oxidative stress injury.
	[238]				EMT	EGCG attenuates EMT in renal tubular cells through GSK-3β/β-catenin/Snail1 and Nrf2 pathways.
	[239]				EMT	EGCG can reverse EMT to mesenchymal-epithelial transition (MET) process in renal cells, to become a potential anti-fibrotic agent to reverse the fibrotic kidney.
The clinical studies of ECGC are lacking, and most evidence was established through animal studies and in vitro studies. The disease focus of the in vitro studies of ECGC mainly focus on EMT and no other type of kidney diseases. To the best of the authors’ knowledge, there was no record regarding to the ECGC studies on autophagy in kidney disease found.						
Resveratrol	Studies of resveratrol autophagy in kidney disease	Animal model	Dosage and route of administration	Combination		
	[240]	35 Male C57BL/6 mice	Cisplatin: 20 mg/kg, single injection. Resveratrol (RES): 30 mg/kg/day intraperitoneally (i.p.). Ginsenoside Rg1 (Rg1): 20 mg/kg/day i.p. Duration: cisplatin: single injection. RES and Rg1 treatments: administered for 3 days before cisplatin injection and continued for another 3 days post cisplatin treatment (6 days in total).	Combined-treatment group received both RES (30 mg/kg/day) and Rg1 (20 mg/kg/day), simultaneously.	Cisplatin induced-AKI	Resveratrol used together with Rg1 alleviated the kidney damage caused by cisplatin, and reduced autophagy was involved in cisplatin-induced AKI
	Clinical trial (Review)					
	[241]					Over the last 20 years, clinical data suggested that resveratrol benefits human health, but high-quality trials needed.
Berberine	Animal studies and in vitro of autophagy in kidney disease	Animal model	Dosage and route of administration	Combination		
	[242]	Male Wistar rats: initial population: 90 rats. Used for high-fat diet and streptozotocin (STZ)-induced diabetic nephropathy model. BKS.Cg-Dock7m+/+Leprdb/JNju (db/db) mice: Total: 40 mice (30 db/db mice and 10 control mice).	Rats: HGSD: 25 mg/kg/day and 100 mg/kg/day orally. Berberine: 100 mg/kg/day orally. Mice: HGSD: 40 mg/kg/day and 160 mg/kg/day orally. Duration: Rats: treatment lasted 16 weeks. Mice: treatment lasted 4 weeks	No combinations were reported in this study. Each treatment group received either HGSD or berberine.	Diabetic nephropathy	High bioavailability of berberine might be connected to the activation of AMPK phosphorylation and protect against diabetic kidney dysfunction.
Abstract access only	[243]	Unspecified rats, in 5 groups. Five groups of rats, number per group not explicitly mentioned: 1. Control (Ctrl). 2. BBR-treated group (no CI-AKI). 3. CI-AKI group. 4. CI-AKI + BBR group. 5. CI-AKI + Tasq (HDAC4 inhibitor) group.	Berberine (BBR): Specific dose not mentioned in the abstract, but is referenced as a treatment group. Ioversol: 10 mL/kg to induce CI-AKI.	CI-AKI + BBR: Evaluating the renal protective effects of berberine alone. CI-AKI + Tasq: testing the effects of HDAC4 inhibition for comparison	Contrast-induced kidney injury	The activation of autophagy-related genes may be associated with berberine and play a protective effect and enhance autophagy.
	[244]-in vitro				High level of glucose induced apoptosis	Berberine alleviating podocyte apoptosis by activating podocyte autophagy.
Rutin	Animal studies of autophagy in kidney disease	Animal model	Dosage and route of administration	Combination		
Abstract access only	[245]	Rats (unspecified in abstract) but groups as follows: 1. Control group. 2. VPA-only group. 3. VPA + RUT (50 mg/kg) group. 4. VPA + RUT (100 mg/kg) group.	Valproic Acid (VPA): 500 mg/kg. Rutin (RUT): 50 mg/kg or 100 mg/kg. Route of administration: not specified in abstract, but likely oral gavage. Duration: 14 days	VPA + RUT (50 mg/kg): to evaluate the protective effects of low-dose RUT. VPA + RUT (100 mg/kg): to evaluate the protective effects of high-dose RUT.	Sodium valproate-induced damage	Rutin treatment protected against kidney damage by attenuating VPA-induced oxidative stress, ER stress, inflammation, apoptosis and autophagy.
Abstract access only	[246]	db/db mice	Rutin: administered at a dose of 200 mg/kg/day, likely oral (not specified in abstract) gavage for 8 weeks.	No other medications were combined with Rutin in the study.	DKD	Rutin restores autophagy through inhibiting HDAC1 via the PI3K/AKT/mTOR pathway in DKD.
The active ingredients or the natural products that are not mentioned in this table but in Table 1 (*Rheum palmatum* L., *Tripterygium wilfordii* Hook F, *Cordyceps sinensis* and *Cordyceps militaris*, *Schisandra sphenanthera*, *Plantago asiatica* and *Plantago major*, *Stephania tetrandra* S Moore, Polyporaceae *Poria cocos* Wolf, Ganoderma, *Silybum marianum*, and *Herba Leonuri*) were lacking in direct evidence and studies conducted to investigate autophagy characteristics in kidney-related diseases.						

## 7. Conclusions

Over the years, there have been significant advances in the investigation of the effects and therapeutic properties of natural products. However, there have been relatively few clinical studies of natural products used in Traditional Chinese Medicine (TCM). This lack of attention to natural products is one of the main reasons for the paucity of clinical research in this area. Phytochemicals, herbs, and other natural products show great promise as directions for future research on the treatment of kidney diseases. Autophagy, a recently discovered process, has garnered significant interest for its association with podocytes. However, the bioavailability of antioxidants in herbs remains a challenge for effective treatment. Some natural products that have shown promise in the treatment of CKD have been linked to autophagy, aging, and cellular senescence. While many natural products have demonstrated positive results, further evidence is needed, particularly about their effects on autophagy, aging, and cellular senescence. At present, it is not recommended to rely solely on natural products for the treatment of kidney diseases. However, the combination of natural products with current clinical therapies may be a promising approach for personalized treatment and for the reduction of adverse effects of mainstream medications. Further research on natural products worldwide could help to deepen our understanding of kidney diseases and their connection to aging.

## Figures and Tables

**Figure 1 ijms-26-00003-f001:**
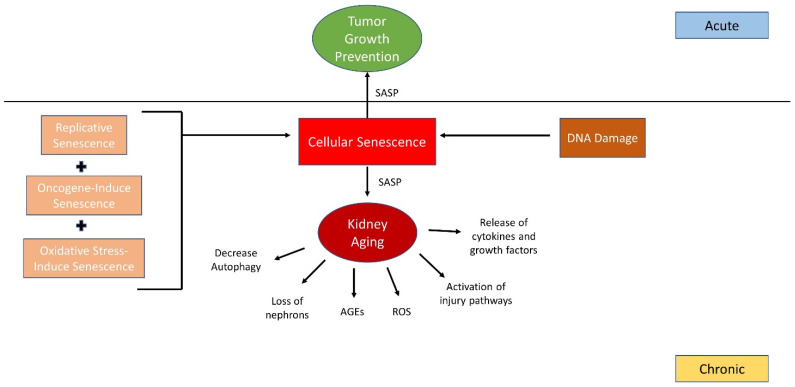
Aging and cellular senescence in kidney aging.

**Figure 2 ijms-26-00003-f002:**
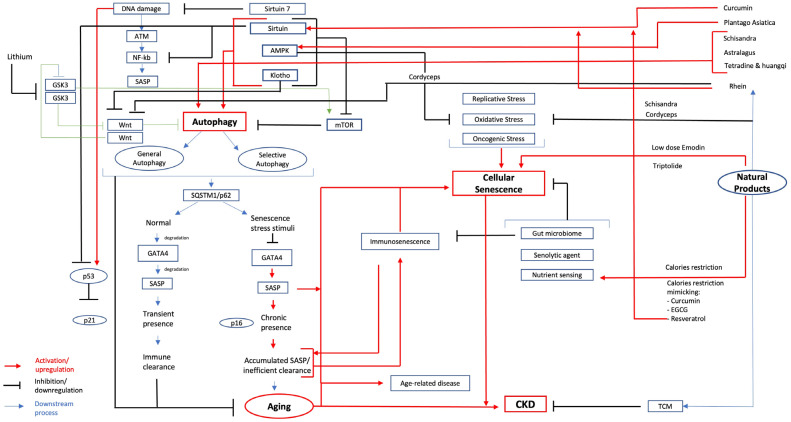
An outline of natural ingredients used in TCM involved in cellular senescence and autophagy. Natural products that can be part of TCM to treat CKD also play their role in activating autophagy and regulating cellular senescence. Curcumin could activate sirtuin and has calorie restriction-mimicking effects that can trigger autophagy and inhibit CS. Hispidulin from *Plantago asiatica* can activate autophagy through AMPK, while Schisandra, Astralagus, Tetradine and Huangqi could activate autophagy. Sirtuin, AMPK, and Klotho could block mTOR, thus activating autophagy. The autophagy mechanism could be affected by the presence of GSK3 and Wnt. Sirtuin 7 could repair DNA damage and prevent it from furthering its downstream mechanism. If the SASP from DNA damage persists, autophagy could be the reliable pathway for performing the SASP clearance. P16 was related to the chronic presence of SASP, while p21 was related to the transient presence of SASP. P53, which could inhibit p21, could be regulated by sirtuin and DNA damage signals. Under senescence stress stimuli-free conditions, the SASP is transient, and will be cleared effectively; unfortunately, if senescence stress stimuli are present, GATA4 would escape from SQSTM1/p62 degradation, causing the buildup amount of SASP, which under chronic presence will cause aging, affecting the adjacent cell which undergoes a similar condition. Ineffective clearance of SASP would cause immunosenescence, which contributes to cellular senescence. The gut microbiome, senolytic agent, and nutrient sensing each have their mechanism of inhibiting cellular senescence. Low-dose emodin and triptolide could induce CS. TCM: Traditional Chinese Medicine, AMPK: 5′AMP-activated protein kinase, mTOR: mammalian target of rapamycin complex, SASP: senescence-associated secretory phenotype, ATM: ataxia–telangiectasia mutate, NF-κβ: Nuclear factor κβ, EGCG: Epigallocatechin-3-gallet, SQSTM1/p62: sequestome 1, GSK3: glycogen synthase kinase 3, Wnt: Wingless-type MMTV integration site, CKD: Chronic Kidney Disease.

**Figure 3 ijms-26-00003-f003:**
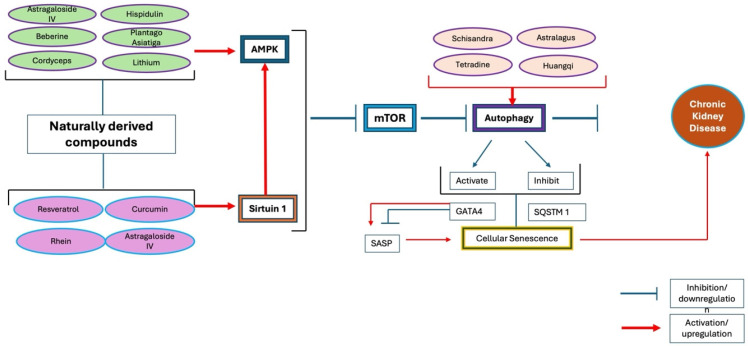
An illustration of Sirtuin, AMPK, and mTOR as the main regulators towards autophagy and cellular senescence that affects chronic kidney disease.

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
