# Peer review of "The Potential of Naturally Derived Compounds for Treating Chronic Kidney Disease: A Review of Autophagy and Cellular Senescence"

_ijms, 2024, doi:10.3390/ijms26010003_

Round 1

Reviewer 1 Report

Comments and Suggestions for Authors

The study titled "The Potential of Natural Derived Compounds for Treating Chronic Kidney Disease: A Review of Autophagy and Cellular Senescence" explores the potential of natural compounds, particularly those used in Traditional Chinese Medicine (TCM), as complementary therapeutic options for chronic kidney disease (CKD). It examines the innovative role of key molecular processes, such as autophagy and cellular senescence, in slowing CKD progression, paving the way for less invasive therapeutic strategies with fewer side effects. Furthermore, the study addresses a critical gap in the literature, providing an analysis of the synergies between traditional and modern treatments, with significant implications for personalized and integrative medicine. However, some suggestions could enhance the quality of the manuscript: 

- I recommend that the authors focus more explicitly on the key findings of the review and clearly state the objectives. The current version is somewhat lengthy and risks losing the reader's attention. Improving the contextualization of the problem and clarifying why natural compounds represent a promising alternative to conventional CKD treatments would also be beneficial. 

- The authors should ensure a smoother transition between sections, clearly explaining how each contributes to the overall topic. 

- Some sections on molecular mechanisms (e.g., cellular senescence and autophagy) are complex and require a more accessible language for a broader scientific audience. 

- Adding illustrative figures or diagrams to better explain the discussed mechanisms, such as molecular pathways involved in autophagy, would enhance clarity. 

- Expand the discussion on the limitations of current studies (e.g., lack of large-scale clinical trials, bioavailability issues of natural compounds). 

- Provide more specific suggestions for future research, such as the need for randomized clinical trials to validate preclinical findings. 

- Improve the analysis of the potential integration of TCM with conventional treatments. Discuss in greater detail the potential risks or interactions between natural compounds and standard medications. 

- Strengthen the conclusions by clearly highlighting the importance of further research on natural compounds. Include a dedicated section on how the findings could be applied to clinical practice.

Comments on the Quality of English Language

The English language could be improved to more clearly express the research.

Author Response

Thank you very much for your valuable comments and suggestions, which have been immensely helpful in improving the manuscript. I deeply appreciate the time and effort the reviewer has dedicated to providing such thoughtful feedback. In response, I have included a table detailing the changes made based on the reviewer’s comments. Additionally, the manuscript is presented in two versions: a clean version and one with tracked changes. The tracked changes version uses red font to indicate added or deleted words and blue font to highlight repositioned paragraphs for easy identification. Each paragraph was number *x* at the beginning of the paragraph for easy search.

Comments

Response

I recommend that the authors focus more explicitly on the key findings of the review and clearly state the objectives. The current version is somewhat lengthy and risks losing the reader's attention. Improving the contextualization of the problem and clarifying why natural compounds represent a promising alternative to conventional CKD treatments would also be beneficial.

Thank you very much for your insightful comments. In response, the authors have repositioned the paragraph to address the issue and clarified why natural compounds represent a promising alternative. This revised section can be found in paragraph 2, highlighted in blue. The updated text is provided below:

Other challenges in studying Traditional Chinese Medicine (TCM) for CKD include the limited availability of English-language studies, formal data documentation, lack of properly designed systematic studies, preclinical studies, clinical studies, and randomized trials. Treatment recommendations are often primarily based on the experience of TCM practitioners [5]. Nonetheless, solid clinical studies have begun to confirm and uncover emerging scientific evidence supporting the mechanisms of TCM [11]. Additionally, certain herbs and ingredients, often passed down by word of mouth, are believed to have kidney-protective effects, which patients may readily adopt in the hope of recovery. Researchers, meanwhile, are increasingly interested in the molecules involved in these protective mechanisms. This review aims to highlight the scientific evidence for kidney-protective effects from natural substances used in TCM, while also providing a perspective on the connections between aging, autophagy, cellular senescence, and these natural substances. The potential involvement of therapeutic targets at the molecular level, not only from TCM but also from other natural products, is discussed to offer researchers direction for further investigation into CKD pathogenesis.

The authors should ensure a smoother transition between sections, clearly explaining how each contributes to the overall topic.

Thank you very much for the valuable suggestion. Another reviewer recommended removing the discussion section from this review paper. In response, the content of the discussion has been redistributed and integrated into the relevant sections to enhance both the clarity and overall coherence of the manuscript.

Some sections on molecular mechanisms (e.g., cellular senescence and autophagy) are complex and require a more accessible language for a broader scientific audience

Thank you very much for the insightful comments. In response, the authors have added more text to provide a detailed explanation of the signaling pathway depicted in Figure 2, ensuring that the description is accessible to a broader audience. Additionally, Figure 3 has been included to illustrate the direct relationship between major autophagy regulators, cellular senescence, and CKD. The added text is located in paragraph 31, while Figure 3 is positioned above paragraph 6. For your convenience, the newly added text has also been provided here for reference:

The metabolites and signaling pathways in Figure 2 offer a comprehensive view of cellular senescence and autophagy, as these pathways are extensive and vary depending on cell type, duration of stressors, and type of stimuli. Figure 2 highlights the relevance of natural products used in CKD management with respect to cellular senescence and autophagy, providing insights into their potential mechanisms for ameliorating CKD. Figure 2 illustrates that under normal, stimuli-free conditions, GATA4 is degraded and regulated by SQSTM1/p62, resulting in a transient presence of SASP and effective immune clearance of senescent cells. In experimental rat models, p21 was found to be transiently elevated, whereas p16 levels increased during sustained cellular senescence [15]. Sirtuin, mTOR, and AMPK in Figure 2 are key regulators of cellular senescence, aging, and various age-related diseases. The role of sirtuins in aging and age-related diseases has been well established in recent studies. The major molecular pathways involved in cellular senescence appeared to be p53, p21, p16, and NF-kb, and sirtuin was reportedly having a direct impact on these molecules [15,128,129]. Figure 2 illustrated the p21 transient presence helps with immune cells clearance hence inhibit aging and the DNA damage is preventing the p21 to clear immune cells. Cellular senescence can occur in both a programmed and a stress-induced manner, involving distinct pathways and processes [36]. Senescence can develop independently through p21 without involving DNA damage response (DDR) and p53, whereas acute and chronic senescence are more complex, engaging various stressors such as p16, DDR activation, p53, and p21 [36]. Consequently, programmed senescence may not lead to cellular senescence accumulation; instead, senescence-related stressors may accumulate as a collateral, detrimental effect. When a certain level of DNA damage is detected, the release of the senescence-associated secretory phenotype (SASP) follows; if SASP persists, it can lead to cellular senescence which later contributes to CKD. The green arrow in Figure 2, indicates the role of GSK3, an enzyme that has two highly conserved isoforms, GSK3α and GSK3β. GSK3β has been reported to directly impact the Wnt signaling pathway, autophagy, and podocyte senescence [37]. Studies suggest that GSK3β acts as an inhibitor of the Wnt signaling pathway (Kreidberg et al., 2022), while conversely, the Wnt signaling pathway can inhibit GSK3β [130]. The Wnt pathway activates mTOR by inhibiting GSK3β; thus, autophagy activation may positively influence cellular senescence. This bidirectional regulation between the Wnt signaling pathway and GSK3β has differential effects on autophagy and cellular senescence requiring further studies to clarify the contradictory role of GSK3. The author believes that exploring the specific underlying conditions related to GSK3 would further enhance our understanding of its role in autophagy.

Adding illustrative figures or diagrams to better explain the discussed mechanisms, such as molecular pathways involved in autophagy, would enhance clarity

Expand the discussion on the limitations of current studies (e.g., lack of large-scale clinical trials, bioavailability issues of natural compounds). 

Thank you for the feedback, the authors have added the limitation of the current studies at paragraph *34 & *35*. The text are added below for the reviewer reference:

*34* For instance, the poor bioavailability and absorption of curcumin have posed significant challenges in fully harnessing its therapeutic potential. However, the use of piperine has been shown to significantly enhance the bioavailability of curcumin [132]. Further studies of this nature are essential to deepen our understanding of naturally derived compounds. Addressing the complexities and unanswered questions surrounding these compounds will bring us closer to their effective application in therapeutic strategies.

*35* Prescribing natural products as supplements alongside standard clinical medications has the potential to enhance treatment outcomes. However, extensive research is still required to substantiate these complementary practices, as many compounds listed in Table 1 have been reported to exhibit toxicity and side effects, including nausea, vomiting, diarrhea, nephrotoxicity, and poor absorption. While certain elements of Traditional Chinese Medicine (TCM) may exhibit toxic properties at specific levels, these characteristics can sometimes stimulate beneficial immune responses that support recovery [133]. However, the effective and safe use of TCM necessitates rigorous standardization, thorough scientific examination, precise dosage measurement, and systematic observation and reporting. Significant gaps remain in the evidence base for these practices, particularly regarding their safety, efficacy, and mechanisms of action. Moreover, due to the experiential nature of TCM—heavily reliant on the expertise and judgment of practitioners—alongside the minor toxic properties used to stimulate immune responses and the variability in individual responses, large-scale clinical trials are essential. Such studies could provide detailed insights into achieving optimal dosage levels while addressing critical issues such as safety and bioavailability. By bridging these gaps, future research could improve the integration of TCM into modern therapeutic strategies, enhancing its safety and efficacy for diverse populations.

Provide more specific suggestions for future research, such as the need for randomized clinical trials to validate preclinical findings.

Improve the analysis of the potential integration of TCM with conventional treatments. Discuss in greater detail the potential risks or interactions between natural compounds and standard medications

Thank you for the valuable feedback. In response, the authors have introduced a new subsection titled Advancing CKD Treatment: Insights into Natural Products, TCM, and Personalized Therapies to thoroughly address the concerns raised.

Reviewer 2 Report

Comments and Suggestions for Authors

In the current review the authors presented the use of natural products in Traditional Chinese Medicine as an alternative treatment for chronic kidney disease and discussed their active ingredients in terms of renoprotection, autophagy, and cellular senescence. A better understanding of the characteristics of these natural products and their effects on autophagy and cellular senescence, coupled with the use of these products as complementary therapy alongside mainstream treatment, may maximize therapeutic benefits and minimize adverse effects for chronic kidney disease patients.

Some suggestions:

1. Add please some statistical data concerning the incidence of chronic kidney disease nowadays.

2. Page 1: The title “Brief view of Traditional Chinese Medicine in CKD” does not correspond to the information presented at pages 1 and 2.

3. Page 3: Give please details concerning the statement “Another interesting fact is that based on large-scale population-based studies, the level of lithium in drinking water significantly correlates with longevity (Fang et al., 2020)”

4. Figure 1 is neither mentioned nor discussed in the article. 5. Page 4: In my opinion you don't have to write about the connection between cellular senescence and tumor growth/cancer. Your article is about  chronic kidney disease. 6. At page 6 you wrote: “The composition of the gut microbiome is associated with various human aging conditions, including cardiovascular disease, cancer, dementia, and other age-related inflammatory processes that contribute to stress-induced macromolecular damage, dysregulation of metabolic activities, and the accumulation of aged immune cells”. Your article is about  chronic kidney disease. Other diseases are not the subject of the article. 7. Pages 6-10: Natural products used in TCM for CKD:

- For each plant/extract/compound you must write first about studies performed on animals and then about the clinical trials. Is mandatory to add the amount of plant/extract/compound administrated and the route of administration. Search please for the plant/extract/compound administrated alone and then the combination of natural products with current clinical therapies.

For animal studies it is not enough to write “animal model”, you must specify the animal. You must also add what is known about the toxicity of the plant/extract/compound. This aspect is very important. At page 13 you wrote: “All the natural substances listed in Table 1 have shown promising results in animal models but lack systematic clinical studies (Table 2)”. I don’t think there are no clinical studies. Please search. 8. Discussion - in my opinion it’s not OK for an review article to have a discussion section - Figure 2 is not properly positioned in the article and is not enough discussed.  

Author Response

The authors sincerely thank you for your thoughtful and constructive feedback. Your comments have been invaluable in improving the manuscript and enriching its content, ensuring it is more informative and beneficial to the audience. In response, the authors have prepared two versions of the revised manuscript: a clean version and a tracked changes version. In the tracked version, changes are highlighted as follows: red font indicates additions and deletions, while blue font marks repositioned paragraphs. Each paragraph has been numbered at the beginning for ease of reference. A detailed response to your comments is provided in the table attached below. Thank you again for your time and effort in reviewing this work.

1. Add please some statistical data concerning the incidence of chronic kidney disease nowadays

Thank you for the valuable suggestion. In response, the authors have included statistics regarding the incidence of CKD, which can now be found in paragraph 1. For the reviewer’s reference, the added text is provided below:

CKD is a significant global health concern, currently affecting approximately 10% of the global population. This statistic underscores the widespread prevalence of the disease [2].

2. Page 1: The title “Brief view of Traditional Chinese Medicine in CKD” does not correspond to the information presented at pages 1 and 2

Thank you for highlighting this issue. In response, the authors have revised the subheading from “Brief View of Traditional Chinese Medicine in CKD” to “Traditional Chinese Medicine in CKD: Aging and Cellular Senescence Focus.” The authors believe this revised subheading better aligns with the content of the text.

3. Page 3: Give please details concerning the statement “Another interesting fact is that based on large-scale population-based studies, the level of lithium in drinking water significantly correlates with longevity (Fang et al., 2020)”

Thank you for the insightful comment. The authors have included additional details to support the statement, which can now be found in paragraph 11. For the reviewer’s reference, the added text is provided below:

Another interesting fact is that based on large-scale population-based studies, the level of lithium in drinking water significantly correlates with longevity [39]. A large Japanese cohort study involving 1,206,174 individuals across 18 municipalities demonstrated a significant inverse correlation between lithium levels in drinking water and all-cause mortality. Similarly, low-dose lithium chloride exposure in Caenorhabditis elegans was shown to extend lifespan (p = 0.047). These findings highlight the potential conserved anti-aging properties of long-term, low-dose lithium exposure across species, emphasizing the need for further investigation into its potential benefits for human longevity [39].

4. Figure 1 is neither mentioned nor discussed in the article

Thank you for highlighting this issue. The authors have addressed it by adding introductory text to accompany the figure in the manuscript. This addition can be found in paragraph 3. For the reviewer’s reference, the text is provided below:

Figure 1 shows cellular senescence contributes to kidney aging through decreased autophagy, loss of nephrons, and activation of injury pathways via senescence-associated secretory phenotype (SASP), while DNA damage together with stress-induced senescence, oncogene-induced senescence and replicative senescence further drive chronic and acute pathological processes.

5. Page 4: In my opinion you don't have to write about the connection between cellular senescence and tumor growth/cancer. Your article is about  chronic kidney disease.

Thank you for the feedback. The authors have revised the manuscript by removing the specified text. The removed content is no longer present in the clean version. For the reviewer’s reference, the removed text can be viewed at: paragraph 4 & 13.

6. At page 6 you wrote: “The composition of the gut microbiome is associated with various human aging conditions, including cardiovascular disease, cancer, dementia, and other age-related inflammatory processes that contribute to stress-induced macromolecular damage, dysregulation of metabolic activities, and the accumulation of aged immune cells”. Your article is about  chronic kidney disease. Other diseases are not the subject of the article.

7. Pages 6-10: Natural products used in TCM for CKD:

- For each plant/extract/compound you must write first about studies performed on animals and then about the clinical trials. Is mandatory to add the amount of plant/extract/compound administrated and the route of administration. Search please for the plant/extract/compound administrated alone and then the combination of natural products with current clinical therapies. For animal studies it is not enough to write “animal model”, you must specify the animal. You must also add what is known about the toxicity of the plant/extract/compound. This aspect is very important. At page 13 you wrote: “All the natural substances listed in Table 1 have shown promising results in animal models but lack systematic clinical studies (Table 2)”. I don’t think there are no clinical studies. Please search.

Thank you for the suggestions, feedback, and comments. The authors deeply appreciate these insights, which have been instrumental in improving the quality of the manuscript. In response, the authors have updated Table 2 to include details on the animal models, dosage, routes of administration, and whether the natural products were used alone or in combination with other medications. Table 2 focuses on autophagy-related studies, highlighting the gap in clinical studies specifically targeting autophagy in CKD, which remains limited. While clinical studies on CKD exist, those with an autophagy focus are scarce. Additionally, Table 1 includes the toxicity and other limitations of natural products as potential drawbacks.

The subheading “Natural Products in TCM” has been organized to present compounds from the most researched to the least. Further elaboration on Tables 1 and 2 can be found in paragraph 32 under the new subheading: Advancing CKD Treatment: Insights into Natural Products, TCM, and Personalized Therapies.

Discussion - in my opinion it’s not OK for a review article to have a discussion section - Figure 2 is not properly positioned in the article and is not enough discussed.

Thank you to the reviewer for highlighting this point. The authors highly value the feedback and have made the necessary revisions by removing the discussion section. The content from the discussion has been redistributed to the relevant paragraphs to enhance clarity and improve the manuscript’s readability. Additionally, Figure 2 has been repositioned and is now discussed in greater detail, which can be found in paragraph 31. For the reviewer’s reference, the newly added text is provided below:

The metabolites and signaling pathways in Figure 2 offer a comprehensive view of cellular senescence and autophagy, as these pathways are extensive and vary depending on cell type, duration of stressors, and type of stimuli. Figure 2 highlights the relevance of natural products used in CKD management with respect to cellular senescence and autophagy, providing insights into their potential mechanisms for ameliorating CKD. Figure 2 illustrates that under normal, stimuli-free conditions, GATA4 is degraded and regulated by SQSTM1/p62, resulting in a transient presence of SASP and effective immune clearance of senescent cells. In experimental rat models, p21 was found to be transiently elevated, whereas p16 levels increased during sustained cellular senescence [15]. Sirtuin, mTOR, and AMPK in Figure 2 are key regulators of cellular senescence, aging, and various age-related diseases. The role of sirtuins in aging and age-related diseases has been well established in recent studies. The major molecular pathways involved in cellular senescence appeared to be p53, p21, p16, and NF-kb, and sirtuin was reportedly having a direct impact on these molecules [15,128,129]. Figure 2 illustrated the p21 transient presence helps with immune cells clearance hence inhibit aging and the DNA damage is preventing the p21 to clear immune cells. Cellular senescence can occur in both a programmed and a stress-induced manner, involving distinct pathways and processes [36]. Senescence can develop independently through p21 without involving DNA damage response (DDR) and p53, whereas acute and chronic senescence are more complex, engaging various stressors such as p16, DDR activation, p53, and p21 [36].